# Soon Filter: Advancing Tiny Neural Architectures for High Throughput Edge Inference

## Abstract

As Deep Neural Networks become more complex and computationally demanding, efficient models for inference at the edge, particularly multiplication-free ones, have gained significant attention. The Ultra Low-Energy Edge Neural Network (ULEEN) is a notable architecture optimized for high throughput edge designs. ULEEN uniquely employs Bloom Filters with binary values to compute neuron activation, boasting better efficiency metrics than Binary Neural Networks (BNNs). This work uncovers a gradient back-propagation bottleneck within ULEEN's Bloom filters and introduces a simplified version of it as a solution: the "Soon Filter". Both theoretically and empirically, we demonstrate that our approach improves gradient back-propagation efficiency. Tests on MLPerf Tiny, MNIST and various UCI datasets reveal that our method surpasses ULEEN, BNN, and DeepShift. Notably, with MLPerf KWS (Key Word Spotting) dataset, we achieve 69.6% accuracy with only 101KiB, while ULEEN, BNN and DeepShift achieve only 67.4%, 55.9%, and 24.9% respectively. Remarkably, we also achieve 67.7% accuracy with only 50KiB, resulting in a 2x model size reduction compared to ULEEN while maintaining similar accuracy (+0.3%). This results underscores the promising potential of our solution for efficient inference at the edge in applications that rely on high throughput architectures.

## 1 Introduction

In recent years, the field of artificial intelligence has witnessed a remarkable transformation due to the advent of Deep Neural Networks (DNNs). These powerful models have pushed the boundaries of what AI can achieve, making significant strides in areas like computer vision (Krizhevsky et al., 2012; Szegedy et al., 2015; He et al., 2016a; Rombach et al., 2021; Ramesh et al., 2021), speech recognition (Hinton et al., 2012; Sainath et al., 2013; MILLET et al., 2022; Li et al., 2022), and natural language processing (Bahdanau et al., 2014; Sutskever et al., 2014; Devlin et al., 2019; OpenAI, 2023). However, this performance comes at a cost, with increasingly complex models demanding higher computational resources. As these models grow in size and complexity, the computational overhead for training and inference becomes substantial, posing a challenge for their deployment in resource-constrained environments.

Inference at the edge, particularly in the growing realm of the Internet of Things (IoT), demands ultra-efficient models. The rapid expansion of the IoT ecosystem has seen an explosion of interconnected devices, from smart thermostats to wearable health monitors. These devices often operate under stringent energy and latency constraints, making it imperative to deploy models that can deliver competitive accuracy without taxing the limited resources available (Shi et al., 2016).

Several optimization techniques, including including Pruning (Dong et al., 2017a;b; Lin et al., 2018), Weight Quantization (Banner et al., 2018; Chmiel et al., 2021; Faghri et al., 2020) and Sparse Neural Networks (Sung et al., 2021; Sun et al., 2021; Ma and Niu, 2018), have been developed, indicating promise in boosting computational efficiency. Nevertheless, while they help in reducing memory consumption, they don't alleviate the inherent computational expenses tied to multiplication operations during the inference stage.

In response, recent research has shifted towards the development of multiplication-free architectures. Binary Neural Networks (BNNs) (Hubara et al., 2016) stand out as a prominent example, in which both weights and activations are quantized to binary values. This paradigm enables the substitution

of multiplication operations with XOR gates, substantially reducing memory and computational overheads (Umuroglu et al., 2017). Concurrently, Deep Shift Networks (Elhoushi et al., 2021a) have been introduced, leveraging shift operations in lieu of multiplications, offering a novel viewpoint on model computational efficiency (You et al., 2020). As a result, multiplication-free models have been deployed in numerous applications (Samragh et al., 2021; Udagawa et al., 2023; Qin et al., 2022; He and Xia, 2018; Xiang et al., 2017; Liu et al., 2021).

In recent advancements in multiplication-free designs, Susskind et al. (2023) introduced the Ultra LowEnergy Edge Neural Network (ULEEN) aimed at applications that rely on high throughput architectures. By utilizing binary-valued bloom filters, along with the use of Straight Through Estimators (STE) and a continuous relaxation of these bloom filters for training, this approach has demonstrated notable improvements in latency, memory consumption, and energy efficiency compared to BNNs, setting new state-of-the-art results and paving the way for the implementation of highly energy-efficient and high throughput models at the edge.

In this work, we theoretically and empirically demonstrate a gradient back-propagation bottleneck present in ULEEN, caused by the use of continuous relaxation of Bloom filters, which hinders learning. Drawing insights and inspiration from ResNet (He et al., 2016a;b) — which emphasizes the benefits of tweaking network architecture to enhance gradient flow — we introduce our solution: the "Soon filter". Both theoretically and empirically, we demonstrate that that our proposed solution ensures more seamless gradient back-propagation to filter locations. Consequently, we set new state-of-the-art benchmarks for multiplication-free high throughput models.

## 2 BACKGROUND

In this section, we provide an overview of the underlying mechanisms of Bloom Filters and ULEEN, which will form the foundation for our methodology.

**Bloom Filter**  The Bloom Filter (Bloom, 1970) is a space-efficient data structure that probabilistically determines membership in a set. It's comprised of a bit array $F \in \{0, 1\}^L$ of fixed size $L$, and $K$ hash functions, denote $h_k$ for each $k \in \{1, \dots, K\}$. Each $h_k$ function maps an element to one of the $L$ positions in the array, indicating its possible presence or absence. Initially, every position in the array is set to 0. When we add an element $e$ to the Bloom Filter, it is hashed, and the corresponding bits in the array change to 1. This is represented as:

$$\forall k \in \{1, \dots, K\} : F_{h_k(e)} \leftarrow 1$$

To verify an element's membership, it's hashed using the same functions, and we check the relevant bits in the array:

$$\theta(e) = \bigwedge_{k=1}^{K} F_{h_k(e)}$$

Here, $\theta(e)$ is the Bloom Filter's output function. If any checked bit is 0, the element isn't in the set. If all are 1, the element might be in the set, but we can't be certain because of potential hash collisions. Hence, the Bloom Filter guarantees true negatives but may produce false positives.

**Straight Through-Estimator**  The Straight Through-Estimator (STE) (Bengio et al., 2013; Yin et al., 2019) is a widely adopted technique for learning binary variables using gradient descent and is commonly employed in BNNs. The STE functions as the sign function during the forward pass and as the derivative of the hardtanh function during the backward pass. This approach allows gradients to pass through the function, which would otherwise be impossible since the derivative of the sign function is infinite at zero and is zero everywhere else. The STE can be expressed as:

$$STE(w) = \begin{cases} 1, & \text{if } w > 0 \\ 0, & \text{otherwise} \end{cases} \qquad \frac{\partial STE(w)}{\partial w} = \begin{cases} 1, & \text{if } |w| < 1 \\ 0, & \text{otherwise} \end{cases}$$

During training, the binary variable to be learned is treated as a real-valued parameter passing through the STE. During inference, as this variable becomes a constant, it is binarized and the STE is removed.

**Weightless Neural Networks**  Weightless Neural Networks (WNNs) are a type of neural model that achieve a multiplication-free characteristic by completely eliminating the use of weights. Instead,

they utilize lookup tables with binary values to determine neural activity, allowing for the deployment of high-throughput models at the edge. Consequently, WNNs have been used in many applications requiring real-time performance (De Gregorio, 2008; Coraggio and De Gregorio, 2007; Do Prado et al., 2007; Carvalho et al., 2014). A drawback of using lookup tables (LUTs) is that the memory requirement grows exponentially with the number of inputs, making the deployment of larger models unfeasible. To address this, (Santiago et al., 2020) proposed substituting LUTs with Bloom Filters, demonstrating that this allows for more efficient and smaller models with negligible changes in accuracy. Additionally, (Susskind et al., 2022) showed that using H3 hash functions (Carter and Wegman, 1979) made the filters extremely efficient for deployment at the edge. Recently, Susskind et al. (2023) further improved upon this work by incorporating gradient-descent training into WNNs using Straight Through Estimators, commonly employed in Binary Neural Networks (BNNs). They also developed a hardware implementation for WNNs named ULEEN, demonstrating its superiority over BNNs in terms of latency, memory usage, and energy efficiency.

**ULEEN** ULEEN serves as a classification model designed to distinguish $C$ distinct classes from a binary input $x \in \{0,1\}^{nN}$. Each class is represented by a discriminator $D_c$ where $c \in \{1, 2, \ldots C\}$. Each discriminator is composed of $N$ Bloom Filters of length of $L$. Every Bloom Filter in the discriminator processes a unique subset of $n$ bits from the input $x$, selected pseudo-randomly. Let $F_{c,i} \in \{0,1\}^L$ denote the bit array of the $i$-th Bloom Filter of the of the $c$-th discriminator. Let $\delta_{c,i,k} \in \{1, 2, \ldots L\}$ represent the hash value of the binary subset produced by the $k$-th hash function, of the $i$ bloom filter, of the $c$-th discriminator. The output of discriminator $D_c$ is expressed by:

$$s_c(x) = \sum_{i=1}^{N} \bigwedge_{k=1}^{K} F_{c,i,\delta_{c,i,k}}$$

where $s_c : \{0,1\}^{Nn} \rightarrow \{1, 2, \ldots, N\}$ indicates the response of the $c$-th discriminator. The discriminator yielding the highest response determines the model's output class. Refer to Figure 1 for a graphical representation.

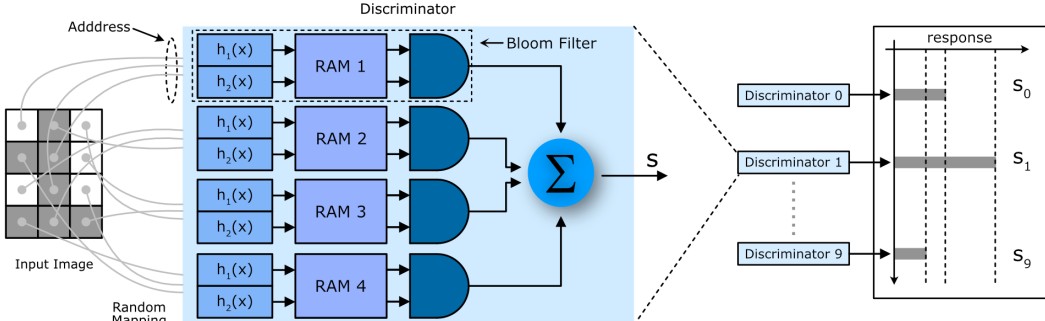

Figure 1: ULEEN doing digit recognition. Each output class has a discriminator. The input image has digit 1 and the discriminator corresponding to digit 1 has the highest response here.

To learn the Bloom Filter's binary values via gradient descent, ULEEN replaces the Bloom Filter AND aggregation with a continuous relaxation—specifically, the min function—during training. The Straight Through-Estimator is employed to learn the filter array's binary values. Specifically, during training, the output of discriminator $D_c$ is expressed as:

$$s_c(x) = \sum_{i=1}^{N} \min_{k=1}^{K} STE(F_{c,i,\delta_{c,i,k}})$$

## 3 SOONFILTER

In this section, we elucidate the inspiration behind our proposed methodology. We theoretically identify a gradient bottleneck in ULEEN's Bloom Filter and introduce our solution: The Soon Filter.

**Inspiration** ResNets (Visin et al., 2015) have shown that altering model architecture to enhance gradient flow during back-propagation can lead to significant improvements in model performance.

Let $f_i$ be the $i$-th layer function in a DNN model. Given a model with $P$ consecutive layers, the partial derivative of the final layer $f_P$ in the with respect to the $i$-th layer $f_i$ can be expressed as:

$$\frac{\partial f_P}{\partial f_i} = \prod_{p=i}^{P-1} \frac{\partial f_{p+1}}{\partial f_p}$$

The authors observed that gradients in earlier layers (those closer to the input) can vanish due to the multiplication of numerous values less than 1 (the partial derivative of one layer with respect to its predecessor) in the chain rule during back-propagation. To mitigate this, they introduced a skip connection between layers. In their new design, assuming a skip connection between every 2 layers, the partial derivative of $f_P$ with respect to $f_i$ became:

$$\frac{\partial f_P}{\partial f_i} = \sum_{u=0}^{P/2-1} \prod_{p=i+2u}^{P-1} \frac{\partial f_{p+1}}{\partial f_p}$$

This architectural alteration ensured that gradients effectively propagated to earlier layers, as elucidated in (He et al., 2016b). This simple yet impactful technique has been integrated into many contemporary DNN architectures (Vaswani et al., 2017; Dosovitskiy et al., 2020; Liu et al., 2022).

Taking cues from this approach of tailoring model architectures to optimize gradient flow, we propose replacing the Bloom Filter in ULEEN with our novel Soon Filter.

**Gradient Bottleneck** Consider $A : \mathbb{R}^K \to \mathbb{R}$, an arbitrary continuous aggregation function. For a loss function $\mathcal{L}$, the partial derivative with respect to an arbitrary content of the ULEEN's Bloom Filter, denoted as $F_{c,i,j}$ (denoting the content at the $j$-th position of the $i$-th Bloom Filter of the $c$-th discriminator) can be written as:

$$\frac{\partial \mathcal{L}}{\partial F_{c,i,j}} = \frac{\partial \mathcal{L}}{\partial A} \frac{\partial A}{\partial STE} \frac{\partial STE}{\partial F_{c,i,j}}$$

We will demonstrate that the middle term of this expression can form a bottleneck, potentially hindering filter positions from updating.

For gradient descent training on ULEEN, a continuous relaxation of the Bloom Filter's AND aggregation function, specifically the min function, is employed. Let $\vec{x} \in \mathbb{R}^K$ be an input representing the output of the STE at the positions accessed by the $K$ hash functions. For the min function, we have:

$$A(\vec{x}) = \min_{k=1}^{K} x_k \qquad\qquad \frac{\partial A}{\partial x_k}(\vec{x}) = \begin{cases} 1, & \text{if } x_k = \min_{k=1}^{K} x_k \\ 0, & \text{otherwise} \end{cases}$$

This function only permits gradients to flow to inputs identical to its own value, effectively blocking gradients to filter positions with values exceeding its own.

Another potential continuous relaxation for the AND aggregation function is the product operation: However, its derivative reveals it to be even more obstructive to gradient back-propagation, resulting in an even greater bottleneck than the min function:

$$A(\vec{x}) = \prod_{k=1}^{K} x_k \qquad\qquad \frac{\partial A}{\partial x_k}(\vec{x}) = \prod_{\substack{l=1 \\ l \neq k}}^{K} x_l$$

In this scenario, the gradient propagates to all filter positions only when the STE for every accessed position yields a 1. If a single STE of a filter position produces a 0, just that particular position is updated by the gradient. When two or more outputs register as zero, none of the filter positions receive a gradient update, effectively halting gradient back-propagation entirely.

**Soon Filter** To overcome this gradient bottleneck issue, we do not constrain ourselves to continuous relaxations of the AND aggregation function. Instead, we choose an aggregation function without a gradient bottleneck:

$$\frac{\partial A}{\partial x_k}(\vec{x}) = 1$$

This function corresponds to the sum operation:

$$A(\vec{x}) = \sum_{k=1}^{K} x_k$$

Using the sum operation as the aggregation function alters the filter's characteristics. In this modified filter, the number of false positives rises. While in the Bloom Filter, indexing both a 0 and a 1 position would produce a true negative, this new filter will output a 1. Due to this filter's tendency to output a result prematurely, we named it the "Soon Filter" — a simplified version of the Bloom Filter that has a higher rate of false positives.

In our approach, the model is trained and deployed exactly like ULEEN but replacing Bloom Filters with Soon Filters. The discriminator's response in our model is as follows:

$$s_c(x) = \sum_{i=1}^{N} \sum_{k=1}^{K} F_{c,i,\delta_{c,i,k}}$$

During training, the discriminator response is given by:

$$s_c(x) = \sum_{i=1}^{N} \sum_{k=1}^{K} STE(F_{c,i,\delta_{c,i,k}})$$

Refer to Figure 2 for a visual representation comparing a discriminator that employs a Bloom Filter with a discriminator that uses a Soon Filter. This comparison illustrates how the increased rate of false positives affects the discriminator's response.

**Analysis of Filter Equivalence** A noteworthy observation arises when examining the Bloom Filter and the Soon Filter within the context of the number of hash functions employed. Specifically, when only one hash function, both the Soon Filter and the Bloom Filter essentially operate as identical filters. This is due to the fact that in the presence of a single hash function, there's no necessity for an aggregation function. Thus the output of both filters is simply the accessed position by that hash function.

Delving deeper into the continuous relaxations of the Bloom Filter, an interesting parallelism can be discerned. For scenarios with one or two hash functions, both continuous relaxations exhibit identical derivatives. This means that, in terms of behavior, the two relaxations are indistinguishable under these conditions. This congruence is evident for a single hash function since it negates the need for an aggregation function. For two hash functions, we can construct a truth table of the partial derivatives to elucidate this:

| Input | | Min | | Product | |
|---|---|---|---|---|---|
| $x_1$ | $x_2$ | $\frac{\partial A}{\partial x_1}$ | $\frac{\partial A}{\partial x_2}$ | $\frac{\partial A}{\partial x_1}$ | $\frac{\partial A}{\partial x_2}$ |
| 0 | 0 | 0 | 0 | 0 | 0 |
| 0 | 1 | 1 | 0 | 1 | 0 |
| 1 | 0 | 0 | 1 | 0 | 1 |
| 1 | 1 | 1 | 1 | 1 | 1 |

When three or more hash functions are introduced, the different continuous relaxations unique characteristics become apparent, highlighting the distinctions in their operational behavior. Below is the truth table showcasing a 3-bit input into the continuous relaxation aggregations functions and their respective derivatives:

| Input | | | Min | | | Product | | |
|---|---|---|---|---|---|---|---|---|
| $x_1$ | $x_2$ | $x_3$ | $\frac{\partial A}{\partial x_1}$ | $\frac{\partial A}{\partial x_2}$ | $\frac{\partial A}{\partial x_3}$ | $\frac{\partial A}{\partial x_1}$ | $\frac{\partial A}{\partial x_2}$ | $\frac{\partial A}{\partial x_3}$ |
| 0 | 0 | 0 | 1 | 1 | 1 | 0 | 0 | 0 |
| 0 | 0 | 1 | 1 | 1 | 0 | 0 | 0 | 0 |
| 0 | 1 | 0 | 1 | 0 | 1 | 0 | 0 | 0 |
| 0 | 1 | 1 | 1 | 0 | 0 | 1 | 0 | 0 |
| 1 | 0 | 0 | 0 | 1 | 1 | 0 | 0 | 0 |
| 1 | 0 | 1 | 0 | 1 | 0 | 0 | 1 | 0 |
| 1 | 1 | 0 | 0 | 0 | 1 | 0 | 0 | 1 |
| 1 | 1 | 1 | 1 | 1 | 1 | 1 | 1 | 1 |

**Hardware Considerations** As depicted, the modifications introduced by our model to the ULEEN hardware are minimal. Specifically, we remove the AND gate from the Filter output and connect the filter outputs directly to the discriminator's pop-count. This ensures that ULEEN's remarkable hardware features and performance remain intact.

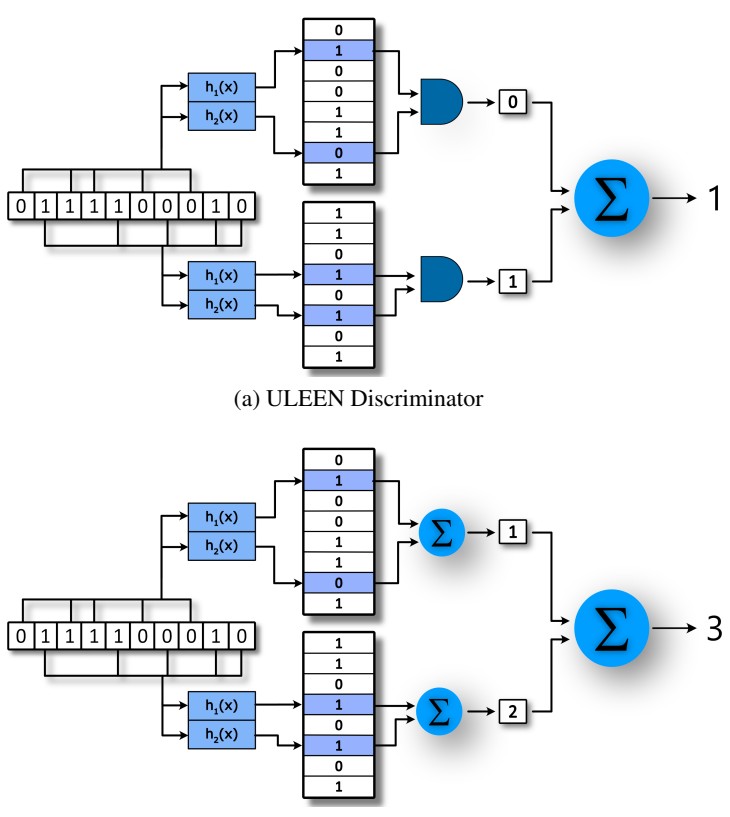

(a) ULEEN Discriminator

(b) SULEEN Discriminator

Figure 2: Graphical Representation of ULEEN (a) and SULEEN (b) discriminators. The two images provide an example of a discriminator that receives a 10-bit input. Each discriminator in the example has two filters that are addressed by two hash functions. Each hash function takes a 5-bit word as input and produces a 3-bit address as output. The distinction between the Bloom Filter and Soon Filter is evident in the images: In (a), the upper Bloom Filter provides a true negative output of 0, while the lower indicates a potential positive, outputting a 1. In (b), the upper Soon Filter outputs a 1 (a false positive) and the lower outputs a 2, indicating a potential positive, totaling an output of 3.

## 4 EXPERIMENTS

In this section, we present the experimental evaluation of our approach, comparing it with ULEEN, BNN, and DeepShift across the MLPerf Tiny benchmark datasets (Banbury et al., 2021), various UCI datasets (Dua and Graff, 2017) and MNIST (Deng, 2012). We designate our model as SULEEN to distinguish it from ULEEN, signifying the incorporation of our proposed Soon Filters within the ULEEN architecture. Moreover, we introduce ablation studies in which we vary the number of hash functions in the Soon Filter and contrast it with the continuous relaxations of the Bloom Filter to corroborate our theoretical filter equivalence conclusions.

### 4.1 EMPIRICAL EVALUATION ACROSS DIVERSE DATASETS

**Model Implementations:** To evaluate BNNs, we utilize FINN (Umuroglu et al., 2017), a specialized tool for developing high-performance neural network architectures on Field-Programmable Gate Arrays (FPGAs). FINN is designed to enable the efficient implementation of BNNs, allowing for significant acceleration in inference compared to traditional architectures. This aligns with the most performant version of BNNs as outlined in the accompanying paper. For DeepShift, we employ the implementation detailed and made available in their paper (Elhoushi et al., 2021a), conducting a grid

Table 1: Accuracy comparison of SULEEN, ULEEN, BNN, and DeepShift Across MLPerf Tiny at three different model sizes (small, medium, and large), various UCI datasets and MNIST. The highest accuracy for each dataset and, for MLPerf Tiny, each model size, is highlighted in bold.

| Dataset | Model Size | SULEEN | ULEEN | BNN | DeepShift |
|---|---|---|---|---|---|
| *MLPerf Tiny* | | | | | |
| KWS(small) | 23KiB | **58.2%** | 57.2% | 47.2% | 18.6% |
| KWS (medium) | 50KiB | **67.7%** | 66.1% | 53.3% | 22.2% |
| KWS (large) | 101KiB | **69.6%** | 67.4% | 55.9% | 24.9% |
| CIFAR-10 (small) | 24KiB | **49.7%** | 45.3% | 40.0% | 40.3% |
| CIFAR-10 (medium) | 250KiB | **55.6%** | 53.5% | 46.5% | 53.0% |
| CIFAR-10 (large) | 625KiB | **57.3%** | 54.5% | 48.0% | 54.1% |
| ToyADMOS (small) | 7KiB | **88.4%** | **88.4%** | 84.8% | 57.8% |
| ToyADMOS (medium) | 15KiB | **89.3%** | **89.3%** | 85.9% | 57.8% |
| ToyADMOS (large) | 30KiB | **90.5%** | **90.5%** | 86.6% | 57.9% |
| VWW (small) | 12KiB | **57.4%** | **57.4%** | 51.7% | 52.9% |
| VWW (medium) | 120KiB | **59.8%** | **59.8%** | 52.1% | 53.8% |
| VWW (large) | 250KiB | **60.6%** | **60.6%** | 52.3% | 54.6% |
| *UCI* | | | | | |
| Ecoli | 0.87KiB | **87.5%** | **87.5%** | 68.9% | 43.6% |
| Iris | 0.28KiB | **98.3%** | 98.0% | 69.2% | 33.3% |
| Letter | 78.00KiB | **96.0%** | 95.3% | 4.79% | 19.2% |
| SatImage | 9.00KiB | **91.7%** | 90.9% | 30.8% | 48.0% |
| Vehicle | 2.25KiB | **78.3%** | 77.1% | 27.2% | 28.3% |
| Vowel | 3.44KiB | **94.0%** | 91.7% | 17.7% | 8.4% |
| Wine | 0.42KiB | **98.3%** | **98.3%** | 14.0% | 27.3% |
| MNIST | (**98**/262/355/408)KiB | **98.6%** | 98.5% | 98.4% | 98.3% |

search on both the Q (quantized) and PS (parameterized shift) variations during hyperparameter tuning. For SULEEN and ULEEN, we developed the code in PyTorch (Paszke et al., 2019), incorporating a custom CUDA kernel. The code is made publicly accessible at: *link omitted due to the double-blind review process*.

**Hyperparameter Tuning:** To optimize each model for every dataset, we employed grid search, utilizing 10% of the training data as a validation set. It is important to note that the test dataset is only used for the final evaluation and not for hyperparameter tuning. For SULEEN and ULEEN, the hyperparameters were varied as follows: $n \in \{2, 3, \ldots, 28\}$, $K \in \{1, 2, 3, 4\}$, and $L \in \{2^1, 2^2, \ldots, 2^n\}$. Common to all models, we explored dropout rates $p \in \{0.0, 0.1, \ldots, 0.8\}$. For BNN and DeepShift, the grid search included the number of hidden layers $P \in \{1, 2, \ldots, 6\}$, with the number of neurons per layer being automatically adjusted to meet the targeted model size. In the case of DeepShift, we additionally conducted a grid search on both the Q and PS variations to discern the optimal choice. Hyperparameter tuning spanned 30 epochs, employing a batch size of 32 and the Adam Optimizer with $\alpha = 0.9$, $\beta = 0.999$. We initiated with a learning rate of 1e-2, reducing it by a factor of 0.1 every 10 epochs.

**Data Splits:** Following the approach in (Susskind et al., 2023), we split the datasets into 66% train and 33% test sets for datasets where no test data was available.

**Preprocessing:** For Keyword Spotting (KWS), we employed Mel Frequency Cepstral Coefficients (MFCC) preprocessing, complemented by cepstral mean and variance normalization. For the UCI datasets and MNIST, we followed the methodology outlined in the ULEEN paper (Susskind et al., 2023).The other datasets underwent no preprocessing. To train both SULEEN and ULEEN models, we employed the Distributive Thermometer Encoding (Bacellar et al., 2022) to binary encode the inputs of all datasets. In the MLPerf Tiny benchmark datasets, we assigned 8 bits for CIFAR-10,

12 bits for KWS, 6 bits for ToyADMOS, and 12 bits for Visual Wake Words (VWW). For the UCI datasets, we utilized a 24-bit encoding across all datasets. For MNIST, we employed a 5-bit encoding.

**Training:** All models was trained for 240 epochs with a batch size of 32. We used the Adam Optimizer with hyperparameters $\alpha = 0.9$ and $\beta = 0.999$. The learning rate was initialized at 1e-2 and decayed by a factor of 0.1 every 80 epochs. Each model was trained and tested 10 times, and we report the average results.

**MLPerf Tiny:** A standard benchmark suite in (Banbury et al., 2021) for edge devices, MLPerf Tiny includes four datasets. Keyword Spotting (KWS) features 105,829 utterances for keyword recognition (Warden, 2018). CIFAR-10 comprises 32x32 RGB images across 10 classes for image classification (Krizhevsky and Hinton, 2009). ToyADMOS/car, with audio recordings of toy cars, focuses on anomaly detection in damaged cars (Koizumi et al., 2019). Visual Wake Words (VWW) uses 96x96 grayscale images from MSCOCO 2014 (Lin et al., 2014) to detect human presence.

**MLPerf Tiny Results:** We conducted experiments across three distinct model sizes (small, medium, and large) for each model and dataset. The results are captured in Table 1. SULEEN consistently excels across all MLPerf Tiny datasets for every model size. Specifically, in the KWS dataset, our large model outperforms DeepShift by 44.7%, BNNs by 13.7%, and ULEEN by 2.2%. Moreover, our medium-sized model (50KiB) achieves comparable accuracy (+0.3%) to ULEEN large model (101KiB), showing an outstanding 2x reduction in memory footprint when compared at iso-accuracy. A similar trend is observed in the CIFAR-10 dataset, where our medium model (250KiB) achieves 55.6%, compared to the ULEEN large model (625KiB) which achieves 54.5%, thereby demonstrating a substantial 2.5x reduction in model size. On ToyADMOS and VWW, SULEEN and ULEEN perform identically. This is due to both achieving optimal results in the hyper-parameter tuning when using a single hash function, causing them to operate essentially as identical models, a phenomenon we detailed theoretically in our methodology and that will be further substantiated in the next subsection.

**UCI:** Our proposed approach is evaluated using a selection of datasets from the UCI Machine Learning Repository (Dua and Graff, 2017). This evaluation aims to verify its applicability for edge inference in applications that utilize structured data.

**UCI Results:** Table 1 encapsulates our findings. SULEEN consistently ranks first in accuracy across all datasets. When juxtaposed against ULEEN, SULEEN exhibits superior accuracy in all cases, save for the Ecoli and Wine datasets, where both models achieve parity. It's striking to note that the top-ranking models in terms of accuracy predominantly belong to WNNs (SULEEN and ULEEN). Both BNN and DeepShift fall short in matching the their performance, accentuating the distinct advantage of WNNs in edge inference applications that utilize structued data.

**MNIST:** In our study, we utilized the MNIST dataset, a classic and foundational benchmark in the field of edge inference. Our focus was on comparing model sizes while maintaining a similar accuracy level, close to 98.5%, a common practice in this domain. We utilize the results reported in (Susskind et al., 2023; Umuroglu et al., 2017; Elhoushi et al., 2021b) for ULEEN, BNN and DeepShift repetively.

**MNIST Results:** The results, as presented in Table 1, demonstrate a significant advancement achieved by SULEEN. Notably, SULEEN attained an impressive accuracy of 98.6% with a model size of only 98KiB. This performance is particularly remarkable when compared to ULEEN, achieving a 2.67x reduction in model size, without compromising on accuracy. Furthermore, when compared to BNN and DeepShift, we acheive a model size reduction of 3.62x and 4.16x respectively. These results underscore SULEEN's significant contribution to the field of edge inference, offering a powerful yet compact solution that does not sacrifice accuracy for size efficiency.

## 4.2 Ablation Studies and Theoretical Validation

In this subsection, we evaluate our Soon Filter against the continuous Min-Relaxation and Prod-Relaxation of the Bloom Filter. We do this by varying the number of hash functions used, ranging from 1 to 4. Our goal is to validate our theoretical findings: that the three filter versions operate as the same model with one hash function and that both the Mean and Product relaxations function identically with two hash functions. Additionally, we aim to confirm that our model consistently outperforms the others regardless of the number of hash functions employed. We test this hypothesis on the Letter, SatImage, Vowel, MNIST, CIFAR-10, and ToyADMOS datasets.

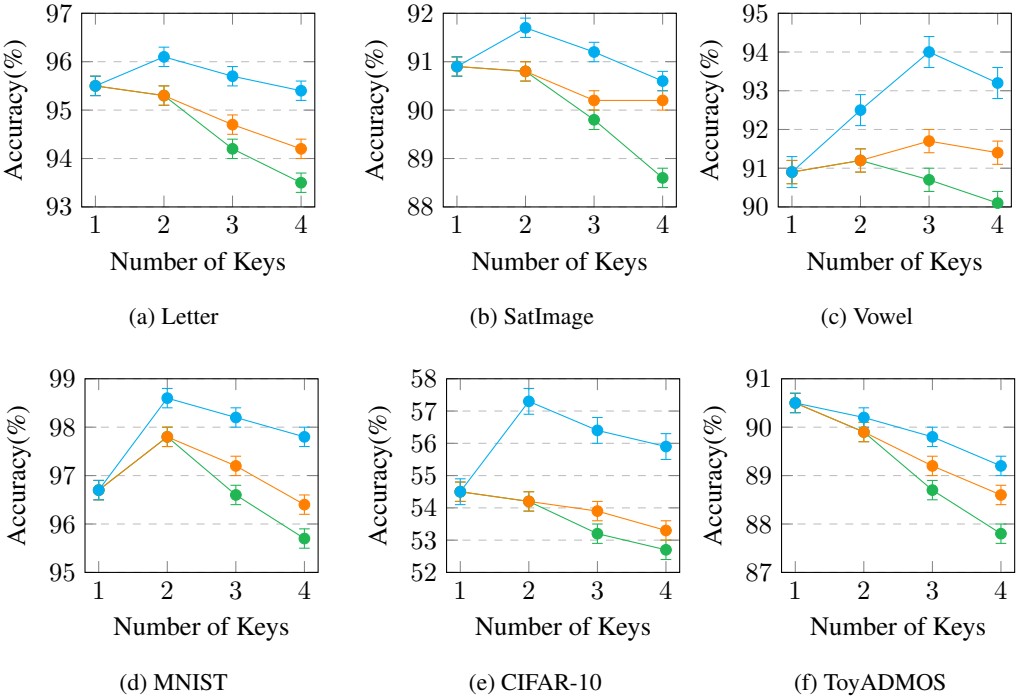

Figure 3: Ablation study graphs comparing the Number of Hash Functions versus Accuracy(%) for the Soon Filter (in Blue), the Bloom Filter with Min function continuous relaxation (in Orange), and the Bloom Filter with Product operation continuous relaxation (in Green). Datasets include (a) Letter, (b) SatImage, (c) Vowel, (d) MNIST, (e) CIFAR-10 , and (f) ToyADMOS .

The results are illustrated in Figure 3. As can be observed, with one hash function, all filters yield identical accuracy across all datasets. When using two hash functions, both Min and Product relaxations also produce identical results. Notably, for every dataset and number of hash functions, the Soon Filter consistently surpasses both the Min and Product relaxations of the Bloom Filter. These findings validate our theoretical assertions.

## 5 CONCLUSION

In this study, we introduced the Soon Filter, an innovative approach designed to enhance the performance of ULEEN, a multiplication-free model tailored for high-throughput edge inference. By theoretically demonstrating its efficiency and conducting rigorous experimentation on the MLPerf Tiny, UCI, and MNIST datasets, we have distinctly underscored the robustness and efficiency of our proposed methodology. Notably, our results have surpassed the performance benchmarks across all datasets, outperforming well-established models like ULEEN, BNN, and DeepShift.

Through ablation studies, we have empirically verified our theoretical assertions regarding filter equivalence, showing that the Soon Filter consistently outperforms its counterparts by maximizing the gradient updates of the filter RAM positions.

Given the minimal deviations between our model and ULEEN, as outlined in the methodology section, we can consider ULEEN's hardware results as a performance upper bound for our model. Additionally, our approach achieves, on average, a 2x reduction in model size compared to ULEEN. Based on this significant size reduction, we project that the hardware deployment of our model could be twice as efficient. This projection underscores the importance of hardware testing as a vital and immediate direction for future research.

Furthermore, numerous edge applications that rely on high-throughput architectures could greatly benefit from our approach. The integration of SULEEN into these applications has the potential to redefine their efficiency and robustness, thereby setting a new standard for edge inference.

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
