# OpenReview forum: "Soon Filter: Advancing Feed-Forward Neural Architectures for Inference at the Edge"
_ICLR.cc/2024/Conference — Submitted to ICLR 2024_

### Official Review · Reviewer_fhaz · 2023-10-26

**Soundness:** 2 fair
**Presentation:** 3 good
**Contribution:** 2 fair
**Rating:** 3
**Confidence:** 4

**Summary:**

This paper proposes a set of technique to realize an implementation of neural networks that are as simple as binary neural networks. Rather than binarization, the authors use hashing functions, inspired from earlier works that have considered filtering techniques. The novelty proposed by the authors is to slightly modify the gradient formulation for back-propagation. A problem with regular STE convergence is identified when using ULEEN, whereby the use of minimum function causes a bottleneck. As an alternative, the authors propose to use the sum function for approximating AND resulting in an always 1 gradient. Experimental results on tiny datasets are provided.

**Strengths:**

The paper is well written, and the exposition of the problem is clear. The presented analysis is clear to follow, and interesting.

**Weaknesses:**

Several issues with this paper can be noted:

-- The main objective of the paper is to reduce the complexity of implementation using hashing and filtering. However, very little is done to measure this complexity. Only the model size is reported, as proxy to complexity. But what about the overhead of implementing hashing and filtering? How does that compare to regular binarization where values are simply clipped to +/-1? How does that compare to low bit width implementation (e.g., 4-bit or 8-bit)? I found the discussion on hardware benefits and limitations of the method very weak.

-- The utilized benchmarks are extremely trivial. The authors have only tested their work on tiny models and datasets. Does the method work on models more relevant to the community in this day and age? E.g., large vision and language models? Even for those tiny datasets, the accuracy is not even close to the state-of-the-art, such as for CIFAR-10 where the authors report accuracies close to 50%.

**Questions:**

Please address questions provided in the above section. Specifically:

-- More nuanced discussion on the hardware implications of the method.

-- More relevant empirical results on typical baselines employed in the ML community in 2023.

---

> ### Author Response · Authors · 2023-11-19
> **Response to Reviewer fhaz**
>
> Thank you for reviewing our paper, providing valuable feedback and helping improve our paper.
>
> **Weakness 1 and Question 1:**
>
> > The main objective of the paper is to reduce the complexity of implementation using hashing and filtering. However, very little is done to measure this complexity. Only the model size is reported, as proxy to complexity. But what about the overhead of implementing hashing and filtering?
>
> We appreciate your feedback and have revised our paper accordingly to clarify its main objective. Contrary to what may have been implied, our goal is not to reduce implementation complexity through hashing and filtering. This aspect was already addressed by the BloomWiSARD paper and further advanced in ULEEN, as detailed in the Weightless Neural Network paragraph of the Background section. ULEEN extensively investigates the overhead of hashing and filtering, demonstrating its superiority over BNN in terms of latency, memory consumption, and energy efficiency. We direct readers to the ULEEN paper for more detailed insights into these aspects.
>
> The primary focus of our paper is to address a gradient bottleneck in the BloomFilter's continuous relaxation. Our proposed solution, the SoonFilter, is detailed in the Methodology Section. It specifically targets this bottleneck, contributing a significant advancement in the field.
>
> > How does that compare to regular binarization where values are simply clipped to +/-1?
>
> Regular binarization, where values are simply clipped to +/- 1, is known as Binary Neural Networks (BNN). We compare against this model thoroughly in the experiments section. Please take a look at Section 4.1 of our revised paper.
>
> > How does that compare to low bit width implementation (e.g., 4-bit or 8-bit)?
>
> The comparison of multiplication-free models with traditional models, as well as with quantized (4-bit, 8-bit, etc) and sparse models, is already well covered in the literature. Therefore, we chose to focus our comparison on the state-of-the-art multiplication-free methods in our experiments. This approach allows us to provide a more direct and relevant assessment of the advancements our Soon Filter brings to this specific category of models.
>
> > I found the discussion on hardware benefits and limitations of the method very weak.
>
> For deployment on edge devices, our work builds upon the ULEEN framework with minimal modifications to its model, as detailed in the methodology section. This strategic choice ensures that our model inherits ULEEN's hardware performance characteristics, which were thoroughly explored in the ULEEN paper. Our model achieves significantly smaller sizes at the same accuracy levels as ULEEN, necessitating fewer filters for comparable accuracy. This efficiency allows us to set ULEEN's hardware performance as an upper bound for our method. Given our inheritance of ULEEN’s hardware characteristics, we chose to focus more on the theoretical development of our proposed method in our paper, as the hardware aspects were already well covered in the ULEEN work.
>
> **Weakness 2 and Question 3:**
>
> In response to your feedback, we have updated our paper to better articulate our benchmark selection. The benchmarks we employed are standard in the edge inference community. This includes MLPerf Tiny, a subset of the MLPerf benchmark suite specifically designed for low-power, edge devices. It focuses on tasks like image classification, keyword spotting, and anomaly detection, crucial for models in resource-constrained environments. MLPerf Tiny offers a standardized framework for consistent and comparable benchmarking across different models and platforms, demonstrating the practical applicability of our approach in edge computing scenarios.
>
> Additionally, we included a collection of UCI datasets, targeting edge applications that use structured data, and included MNIST as it is consistently used in all other papers we compare against, ensuring a relevant and direct comparison. The results for comparison models on MNIST are sourced directly from their respective papers.
>
> For a detailed explanation of our benchmark choices, please refer to the Subsection 4.1 of our revised paper.
>
> We hope that these updates and clarifications adequately address your concerns. If you have any further issues or need additional clarification, we are more than happy to assist or make necessary revisions as needed.

---

> > ### Comment · Reviewer_fhaz · 2023-11-20
> > **Thanks for the response**
> >
> > I acknowledge having read the response by the authors. I am keeping my original review and score.

---

### Official Review · Reviewer_aeHv · 2023-10-30

**Soundness:** 2 fair
**Presentation:** 3 good
**Contribution:** 1 poor
**Rating:** 3
**Confidence:** 5

**Summary:**

This paper presents a multiplication-free neural network based on bloom filters, making it suitable for deployment on ultra-constrained edge/IOT devices. In fact, the paper focuses on a specific optimization to bloom filters and prior work (ULEEN) by replacing a non-differentiable logic function (AND) with a summation operation. This helps gradient flow and trainability making the NN achieve higher accuracy on an assortment of tasks.

**Strengths:**

The paper is clearly-written and well-motivated and the empirical improvements are consistent on a number of tasks.

**Weaknesses:**

This paper is interesting and I learned something new about multiplication-free circuits. However, the proposed modification is very small (compared to ULEEN), the application is quite niche and not well demonstrated, the improvements are very small, and the evaluation is inadequate. This work is promising but there are more questions that need to be answered to provide a compelling argument for the presented approach.

**Questions:**

- How does your performance compare to CNNs?
- How general is your approach? Can it be applied to CNNs, attention, other tasks, other NN sizes?
- DId you deploy this on a challenging edge device? how does end-to-end performance compare to alternatives?

---

> ### Author Response · Authors · 2023-11-19
> **Response to Reviewer aeHv**
>
> Thank you for reviewing our paper, providing valuable feedback and helping improve our paper.
>
> **Weakness:** Concerning the perceived smallness of our contribution, we understand that replacing the AND operation with a SUM in Bloom Filters might seem minor. However, we argue that this is a non-trivial change. Such a substitution is counterintuitive in Bloom Filters as it increases the false positive rate, which is typically undesirable. This alteration is grounded in a robust theoretical framework of gradient maximization, as elaborated in our methodology section. Our work, drawing inspiration from the developments in resnets, mirrors a similar pattern. At first glance, adding a skip connection in resnets might also appear a small contribution. Yet, their work became a cornerstone for many modern deep neural network (DNN) models. We contend that our contribution is significant and impactful, as evidenced by the results we achieved. Notably, our Soon Filter consistently halves the model size of ULEEN, marking a substantial advancement in multiplication-free models. For comprehensive evidence, please refer to the **MLPerf Tiny Results** and **MNIST Results** paragraphs in the experiments section in the revised paper. This size reduction facilitates the deployment of much more efficient models at the edge.
>
> **Questions 1 and 2:** Weightless neural networks (WNNs) hold significant potential for operating with extremely low computational resources. However, they encounter a major challenge that must be addressed to make them applicable in more complex scenarios: their inability to achieve greater depth due to the discrete nature of the RAM indexing, which lacks a derivative, akin to embeddings. As a result, constructing deep models and convolutional models with WNNs is currently unfeasible. This limitation presents a substantial opportunity for future research. Overcoming this challenge could lead to groundbreaking advancements in deep learning, paving the way for the development of even more efficient and effective models. Consequently, future research efforts should concentrate on devising algorithms capable of facilitating back-propagation with discrete operations such as RAM indexing.
>
> **Question 3:** For deployment on edge devices, our work builds upon the ULEEN framework with minimal modifications to its model, as detailed in the methodology section. This strategic choice ensures that our model inherits ULEEN's hardware performance characteristics, which were thoroughly explored in the ULEEN paper. Our model achieves significantly smaller sizes at the same accuracy levels as ULEEN, necessitating fewer filters for comparable accuracy. This efficiency allows us to set ULEEN's hardware performance as an upper bound for our method. Given our inheritance of ULEEN’s hardware characteristics, we chose to focus more on the theoretical development of our proposed method in our paper, as the hardware aspects were already well covered in the ULEEN work.
>
> We hope that these updates and clarifications adequately address your concerns. If you have any further issues or need additional clarification, we are more than happy to assist or make necessary revisions as needed.

---

> > ### Comment · Reviewer_aeHv · 2023-11-21
> > **keeping my score**
> >
> > Thanks for your response.
> >
> > I am keeping my score. While the author's comments make sense, I am not convinced that replacing AND with SUM is as impactful as skip connections in ResNet. Skip connections were key to gradient flow and training deeper CNNs thus unlocking a huge design space. In the case of Uleen and the presented work, the impact is much smaller and the evaluation still lacks depth, and the contributions (while solid) are somewhat minor, warranting more work before publication in this reviewer's opinion.

---

### Official Review · Reviewer_N6Ny · 2023-11-01

**Soundness:** 3 good
**Presentation:** 3 good
**Contribution:** 2 fair
**Rating:** 5
**Confidence:** 3

**Summary:**

This study primarily aims to reveal a noteworthy bottleneck in the gradient back-propagation process of ULEEN. This bottleneck is attributed to the continuous relaxation of Bloom filters, which hinders the learning process. Drawing inspiration from the principles emphasized in ResNet, which stress the benefits of fine-tuning network architecture to enhance gradient flow, this study introduces a solution known as the "Soon filter." Theoretical and empirical results demonstrate that this proposed solution substantially enhances the smooth back-propagation of gradients to filter locations. Consequently, it establishes a new state-of-the-art benchmark for multiplication-free feed-forward models.

**Strengths:**

1. The paper is well-structured and easy to follow.

2. The primary goal is to set a new state-of-the-art benchmark for multiplication-free feed-forward models.

3. The introduction of the "Soon filter" addresses the gradient flow issue.

**Weaknesses:**

1. Incorporating additional visual aids and illustrations within the section on related work would greatly enhance the clarity and comprehensibility of the paper. These visuals can provide readers with a more comprehensive understanding of the prior research landscape in the field.

2. While the central contribution of the paper revolves around the introduction of the "Soon Filter," it is essential to consider augmenting this with supplementary contributions or by providing more extensive exploration and insights to enrich the overall content and scholarly impact.

**Questions:**

1. Based on Figure 1 and the main content, is the primary distinction between these methods solely attributable to the replacement of the OR operation with the SUM operation?

2. Regarding the VWW dataset, have you evaluated the performance of these methods on smaller models like MCUNetV1 [1] and MCUNetV2 [2]? Such an analysis could provide insights into the accuracy reduction when transitioning from multiplication-based models and offer valuable insights.


[1] MCUNet: Tiny Deep Learning on IoT Devices

[2] MCUNetV2: Memory-Efficient Patch-based Inference for Tiny Deep Learning

---

> ### Author Response · Authors · 2023-11-19
> **Response to Reviewer N6Ny**
>
> Thank you for reviewing our paper, providing valuable feedback and helping improve our paper.
>
> **Weakness 1:** In response to your observation, we have added a new Figure in our manuscript. The new figure now offers a more detailed and descriptive visual representation, enhancing the clarity and understanding of the prior work. We believe this adjustment will significantly improve readers' comprehension of the context and background of our study.
>
> **Weakness 2:** Our primary focus is indeed on the introduction of the Soon Filter. We argue that this represents a high-impact contribution, as demonstrated by the results we have obtained. Notably, our Soon Filter consistently reduces the model size of ULEEN by half, a significant advancement. For detailed evidence of this, please refer to the **MLPerf Tiny Results** and **MNIST Results** paragraphs in the experiments section in the revised paper. This reduction enables the deployment of far more efficient models at the edge, a contribution we believe to be of considerable scholarly significance.
>
> **Question 1:** Yes, the distinction between the Bloom Filter and the Soon Filter is solely attributable to the substitution of the AND operation with the SUM operation. This key difference is thoroughly detailed in our methodology section. Furthermore, in our experiments, we have ensured that the training regime for both ULEEN and our proposed SULEEN remains the same, with the only difference being the use of the Bloom Filter and the Soon Filter, respectively. Please refer to the  **Hyperparameter Tuning**, **Preprocessing**, and **Training** paragraphs in the experiments section in the revised version of our paper for further details.
>
> **Question 2:** The comparison of multiplication-free models with traditional models, as well as with quantized and sparse models, is already well covered in the literature. Therefore, we chose to focus our comparison on the state-of-the-art multiplication-free methods in our experiments. This approach allows us to provide a more direct and relevant assessment of the advancements our Soon Filter brings to this specific category of models.
>
> We hope that these updates and clarifications adequately address your concerns. If you have any further issues or need additional clarification, we are more than happy to assist or make necessary revisions as needed.

---

### Author Response · Authors · 2023-11-19
**Dear Review Committee**

Dear Review Committee,

First and foremost, we wish to extend our sincere gratitude for your thorough and insightful reviews of our paper. Your feedback has been invaluable in guiding our revisions, and we have made concerted efforts to address the points raised.

We have revised and updated our manuscript in several key areas in response to the feedback.

We hope that these revisions adequately address the concerns raised. If there are any further issues or aspects that you feel have not been fully addressed, we would be more than happy to provide additional clarification or make further revisions as needed.

---

### Meta-Review · Area_Chair_pmAG · 2023-12-11

**Metareview:**

The paper is concerned with learning multiplication-free neural networks for low resource edge devices and uses bloom filters to achieve this  (based on prior work ULEEN). The main contribution of the paper to modify the prior work by replacing the non-differentiable logical AND with a summation that makes it more amenable to gradient based optimization. Reviewers think the proposed modification is small and the evaluation isn't thorough enough across different architectures and model sizes. The paper isn't suitable for publication at ICLR in its current form but future versions might benefit from adding more empirical results that convincingly demonstrate the advantages of the method, which can help make up for the lack in technical content.

**Justification For Why Not Higher Score:**

Insufficient technical contribution; lack of comprehensive evaluations across models.

**Justification For Why Not Lower Score:**

N/A

---

### Decision · Program_Chairs · 2024-01-16

Reject